# Antimicrobial Activity of Biogenic Metal Oxide Nanoparticles and Their Synergistic Effect on Clinical Pathogens

**DOI:** 10.3390/ijms24129998

**Published:** 2023-06-10

**Authors:** Dali Vilma Francis, Manju Nidagodu Jayakumar, Hafiz Ahmad, Trupti Gokhale

**Affiliations:** 1Department of Biotechnology, Birla Institute of Technology and Science, Pilani, Dubai Campus, Dubai International Academic City, Dubai P.O. Box 345055, United Arab Emirates; p20170903@dubai.bits-pilani.ac.in (D.V.F.); p20190908@dubai.bits-pilani.ac.in (M.N.J.); 2Research Institute for Medical and Health Sciences, University of Sharjah, Sharjah P.O. Box 27272, United Arab Emirates; 3Department of Medical Microbiology & Immunology, RAK College of Medical Sciences, RAK Medical and Health Sciences University, Ras Al Khaimah P.O. Box 12973, United Arab Emirates; hafiz@rakmhsu.ac.ae; 4Adjunct Clinical Microbiologist and Head of Molecular Division, RAK Hospital, Ras Al Khaimah P.O. Box 12973, United Arab Emirates

**Keywords:** antimicrobial activity, metal oxides, CuO nanoparticle, ZnO nanoparticle, WO_3_ nanoparticle, synergistic action, lipid peroxidation, flow cytometry

## Abstract

The rising prevalence of antibiotic-resistance is currently a grave issue; hence, novel antimicrobial agents are being explored and developed to address infections resulting from multiple drug-resistant pathogens. Biogenic CuO, ZnO, and WO_3_ nanoparticles can be considered as such agents. Clinical isolates of *E. coli, S. aureus*, methicillin-resistant *S. aureus* (MRSA), and *Candida albicans* from oral and vaginal samples were treated with single and combination metal nanoparticles incubated under dark and light conditions to understand the synergistic effect of the nanoparticles and their photocatalytic antimicrobial activity. Biogenic CuO and ZnO nanoparticles exhibited significant antimicrobial effects under dark incubation which did not alter on photoactivation. However, photoactivated WO_3_ nanoparticles significantly reduced the number of viable cells by 75% for all the test organisms, thus proving to be a promising antimicrobial agent. Combinations of CuO, ZnO, and WO_3_ nanoparticles demonstrated synergistic action as a significant increase in their antimicrobial property (>90%) was observed compared to the action of single elemental nanoparticles. The mechanism of the antimicrobial action of metal nanoparticles both in combination and in isolation was assessed with respect to lipid peroxidation due to ROS (reactive oxygen species) generation by measuring malondialdehyde (MDA) production, and the damage to cell integrity using live/dead staining and quantitating with the use of flow cytometry and fluorescence microscopy.

## 1. Introduction

The emergence and spread of multidrug-resistant (MDR) pathogens represent a grave danger to healthcare worldwide [1], as antimicrobial treatments are significantly imperiled [2] and offer only a temporary solution as the pathogens continue to evolve and exhibit a greater resistance. Therefore, alternative strategies are required to tackle these pathogens. Non-toxic and noninvasive approaches are being explored by the healthcare industries to control and prevent infection. Designing and employing non-toxic antimicrobial biomaterials that can be used directly or indirectly in the healthcare system (i.e., surfaces in the manufacturing of drugs and medical devices), may obviate the use of conventional antimicrobial agents [3]. 

Various metal nanoparticles have been reported to possess antimicrobial properties [4], and hence can be used in medical and pharmaceutical devices to prevent the spread of multi-drug resistant infections in the healthcare industry [5]. Incorporation of these antimicrobial nanoparticles into the clinical and industrial devices offers a novel, cost-effective solution, which also lowers the consumption of disinfectants and the involvement of disinfecting processes [6]. Biogenic silver nanoparticles have exhibited a promising antimicrobial activity which has extended research on the applications of various other nanoparticles in healthcare [7]. Similar to antibiotics, different chemically synthesized metal nanoparticles of gold, silver, copper, zinc, and iron have exhibited a varied effect on microbial pathogens [5], and hence it is essential to explore the potential of different biogenic metal nanoparticles or their combinations to treat drug-resistant microbes [8]. Metal nanoparticles exhibit a non-specific antimicrobial activity and can employ mechanisms such as the generation of pores on the cell wall or surface membrane, ROS generation, or interference in DNA replication or gene expression. Each metal nanoparticle may exhibit a different mode of action, and hence combinations of metal nanoparticles could use multiple mechanisms to combat a broad range of microorganisms [9,10]. Therefore, the integration of nanoparticle combinations could be a viable option for treating medical devices or surgical platforms to prevent infections [11,12]. Some metal nanoparticles such as titanium dioxide, gold, silver, and tungsten have been reported to possess photocatalytic activity [13], however the application of their photocatalytic property to destroy microorganisms is hitherto unexplored. 

*Staphylococcus aureus*, methicillin-resistant *Staphylococcus aureus* (MRSA), *E coli*, and *Candida albicans* are examples of predominant pathogens, as they are highly resistant to a couple of antibiotics [14,15]. They also constitute up to 50% of nosocomial infections due to their ability to form biofilms on equipment which could lead to respiratory infections such as ventilator-associated pneumonia in prolonged ICU admitted patients or immunocompromised patients. Development of biofilms lower the equipment effectiveness and increase the risk of equipment corrosion [15,16].

The aim of this study was to evaluate the antimicrobial activity of biogenic CuO, ZnO, and WO_3_ nanoparticles and their combinations against clinical isolates, study the photocatalytic effect of these nanoparticles on their antimicrobial activity, and reveal the mechanism of action of the metal nanoparticles in bacteria and yeast. 

## 2. Results

### 2.1. Synthesis of CuO, ZnO, and WO_3_ Nanoparticles

The biogenic synthesis of CuO, ZnO, and WO_3_ nanoparticles was initially verified by particle size analysis on the Malvern Zetasizer Nano ZS particle size analyzer. The particle size distribution of water-suspended CuO, ZnO, and WO_3_ nanoparticles is presented in Figure 1. The average particle size of 81.23 nm, 89.79 nm, and 90.18 nm was obtained for CuO, ZnO, and WO_3_ nanoparticles, respectively. The biosynthesized nanoparticles were lyophilized and then considered for further characterization. 

### 2.2. Characterization of the Nanoparticles

The XRD (Appendix A) and ATR-FTIR (Appendix A) spectra confirmed the biosynthesis of CuO, ZnO, and WO_3_ nanoparticles. The diffraction peaks at 2θ values of 32.3795, 35.7643, 38.8219, 46.2562, 48.7309, 53.4758, 58.4047, 61.5645, 66.7287, and 68.191 degrees, corresponding to (111), (000), (200), (−112), (−202), (020), (202), (−113), (022), and (220) planes, respectively, confirmed the presence of copper oxide, and 2θ values of 31.94, 34.64, 36.42, 47.83, 56.85, 62.93, and 68.2 (100, 002, 101, 102, 110, 103, and 112) were due to the zinc oxide diffraction. The XRD patterns of the lyophilized nanoparticles displayed a reflection index as (002), (020), (200), (202), (222), (321), (140), and (−114), respectively. The XRD patterns (Appendix A), in comparison with the standards (JPCDS card No: 83-0950) established the presence of WO_3_ nanoparticles in the lyophilized powder. Using the Debye–Scherrer equation, the average grain size of the CuO, ZnO, and WO_3_ nanoparticles were determined as 38.96 (± 5) nm, 42.64 (±3) nm, and 53.03 (±5) nm, respectively (Table 1). 

ATR-FTIR analysis (Appendix A) of the lyophilized nanoparticles demonstrated a few prominent absorption peaks at 1662 cm^−1^, 1558 cm^−1^, 1392 cm^−1^, 1000 cm^−1^, 1073 cm^−1^, 803 cm^−1^, and 602 cm^−1^, respectively. The peaks at 602 cm^−1^ and 803 cm^−1^ were found to be associated with O–W–O stretching and W–O–W bending vibrations, while the peak at 1000 cm^−1^ was attributed to W=O stretching, thereby suggesting the biosynthesis of WO_3_ nanoparticles. In Appendix A, the FTIR peaks at 433 cm^−1^ correspond to the typical stretching vibration of the Cu-O bond, while in Appendix A, the FTIR peaks at 575 cm^−1^ were related to Zn-O stretching. The appearance of bands with the wavenumbers of 1625 and 1550 cm^−1^ correspond to the bending vibrations of the protein amides I and II, respectively. The peaks around 3450 cm^−1^ have been attributed to the stretching vibrations of amide I superimposed on the side of the hydroxyl group band, which supports the existence of the proteins and, consequently, the biosynthesis of WO_3_, CuO, and ZnO nanoparticles. SEM analysis revealed that the morphology of CuO, ZnO, and WO_3_ nanoparticles was spherical and less than 100 nm (Appendix A) [17,18,19].

### 2.3. Minimum Inhibitory Concentration (MIC) by Resazurin Assay

The effect of the action of metal nanoparticles on the viability of the test organism was assessed using the resazurin assay [20]. As shown in Table 2, CuO nanoparticles exhibited maximum antimicrobial activity against all the test microorganisms used in the study, followed by ZnO nanoparticles, while WO_3_ nanoparticles exhibited the least activity. CuO nanoparticles were the most effective against *Candida albicans* (V) with a MIC at 15.6 ng/µL and were the least effective against MRSA and *E. coli*, with a MIC at 62.5 ng/µL. ZnO nanoparticles were the most effective against *Candida albicans* (O) and *Candida albicans* (V) with a MIC at 31.25 ng/mL, while MRSA and *E. coli* were resistant at higher concentrations (Figure 2). It was immensely disappointing to observe the resistance rendered by all the test microorganisms towards WO_3_ nanoparticles except for *S. aureus*, which was inhibited at 250 ng/µL. 

The WO_3_ nanoparticles were hence exposed to light for studying their potential photocatalytic antimicrobial activity [21]. The exposure of the WO_3_ nanoparticles to visible light surprisingly enhanced their antimicrobial activity by several folds, thereby reducing the MIC to 15.6 ng/mL or lower for all the test microorganisms (Figure 2). However, there was no significant boost in the antimicrobial properties of CuO and ZnO nanoparticles upon their exposure to visible light (Figure 2) and exhibited no significant changes in their activity. 

### 2.4. Minimum Microbicidal Concentration (MMC)

Minimum microbicidal concentration of the nanoparticles deviated slightly from the pattern observed for the MICs. *C. albicans* (V) was he most sensitive towards 15.6 ng/mL CuO nanoparticles incubated in the dark, while all cultures were found to be resistant towards the antimicrobial effect of WO_3_ nanoparticles in the dark. However, the MMC of the WO_3_ nanoparticles reduced significantly upon the shining of the light, a result which was similar to that observed for the MIC. A surprising result observed was the increased anti-microbicidal effect of the ZnO nanoparticles against *E. coli* cells, where the MMC dropped from 250 ng/mL to 31.25 ng/mL, respectively (Table 2). 

### 2.5. Determination of Malondialdehyde (MDA)

The amount of MDA produced due to the damaging action of the nanoparticles was found to be significantly higher than that recorded for all the negative controls, i.e., ≤1.5 µM for both dark and light incubation treatments (Figure 3). The amount of MDA produced in the organisms treated with the CuO and ZnO nanoparticles was >2.5 µM, which was found to be significantly higher (0.05 > *p*) than the negative controls. However, there was negligible or least significant differences (0.05 < *p*) observed in the MDA production among the test organisms under light and dark incubation with the CuO and ZnO nanoparticles. The treatment with WO_3_ nanoparticles demonstrated a surprising effect, where the test organisms (except for *E. coli*), when incubated in the dark, produced a marginally higher amount of MDA, though this was deemed to be non-significant (0.05 ≥ *p*) compared to the controls, whereas *E. coli* produced significantly higher amounts of 2.32 µM MDA. Changing the incubation in the presence of light witnessed a surge in the production of MDA (>3 µM) in all test organisms treated with WO_3_ nanoparticles, which was found to be significantly higher (0.05 > *p*) than that recorded under dark treatment (Figure 3), thereby, reflecting the enhanced potential of WO_3_ nanoparticles as inhibitory agents upon photoactivation. Furthermore, this suggests that all test organisms treated with WO_3_ nanoparticles in the light would produce more ROS than those treated with WO_3_ nanoparticles under dark incubation. 

The test organisms treated with different combinations of metal nanoparticles exhibited a significant increase in MDA content as compared to the single elemental nanoparticle treatments.

The presence of WO_3_ nanoparticles in all combinations stemmed a positive antimicrobial effect under photoactivation (Figure 3), leading to an enhanced MDA production.

The boxplot in Figure 4 represents the mean concentration of MDA produced by all the test organisms when treated with CuO, ZnO, and WO_3_ nanoparticles and their combinations C1, C2, and C3 under light and dark incubation. The light-incubated C2 combination of nanoparticles expressed a subtle increase in MDA content in a narrow range among all test organisms. On the other hand, WO_3_ nanoparticles under dark treatment exhibited a wide range of MDA production in the test organisms. 

All pathogenic cultures treated with elemental oxide nanoparticles and in combination with or without light incubation exhibited an increase in lipid peroxidation. Since lipids are vital macromolecules for the microbial cell membrane, the Live/Dead BacLight Bacterial Viability Kits were used to assess the membrane integrity and viability of these cultures.

### 2.6. Flow Cytometry Analysis of Dead/Live Microbial Cells and Membrane Integrity Assay by Fluorescence Microscopy Imaging

The test organisms exposed to different metal nanoparticles in singles and combinations were stained with a BacLight kit to determine their membrane integrity. The staining created a distinct and repeatable fluorescence pattern that was able to be measured using flow cytometry. Flow cytometry analysis of a known test sample that contained an equal number of live and heat-killed *E. coli* cells stained with the BacLight kit demonstrated an equal distribution of *E. coli* cells stained with SYTO9 and PI, respectively, without any significant differences observed (*p* value > 0.05) (Figure 5). A significant difference in the signal of two log decades was observed for the two populations without any apparent variation. This confirmed the reliability of the selected staining and analysis method. 

Flowcytometric analysis of an *E. coli* positive control revealed 99% of live cells (Figure 6), while the other test organisms expressed a maximum viability of 95–100% prior to their exposure to the metal nanoparticle. The positive controls did not exhibit any significant variation in their cell viability upon their incubation in the dark and in the presence of light, and hence the presence or absence of light was found to not influence the cell viability. CuO nanoparticles exhibited a significantly higher antimicrobial activity against the five selected test organisms when compared to the other nanoparticles incubated in the dark, while *E. coli* was found to be the most sensitive among the test organisms (Figure 6). It was surprising to observe the resistance exhibited by all test organisms towards WO_3_ nanoparticles under dark incubation, where only up to 21% of *E. coli* cells were affected. However, on incubating in the presence of light, WO_3_ nanoparticles exhibited a surge in their antimicrobial activity, and 82.7% of *E. coli* cells were affected as a result and were thus stained dead. More than 70% of cell death was observed in all five cultures treated with WO_3_ nanoparticles in the presence of light with maximum cell death observed in *E. coli* (82.7%), followed by *S. aureus* (81.2%), and MRSA (78.3%). A similar effect of enhanced cytotoxicity under light incubation was observed for ZnO nanoparticles in MRSA. ZnO nanoparticles incubated in the dark affected the viability of 45.6% of MRSA cells, while under light incubation affected 73.7% of MRSA cells. A significant effect (*p* value < 0.05) on the MRSA cells under light incubation was deemed as remarkable, as other test organisms lacked any significant deviation (*p* value > 0.05). Except for MRSA, incubation in the presence of light made no significant changes in the antimicrobial activities of CuO and ZnO nanoparticles (*p* value > 0.05). 

All the nanoparticle combinations were found to be equally effective against the five selected test organisms under dark incubation (>80%). Each nanoparticle, when used independently, did not affect the viability of the test organisms, but all combinations exhibited a synergistic effect resulting in cell death. The cytotoxic effect of the nanoparticle combinations increased marginally (>90%) but was not significantly high upon exposure to light. The microscopic images of the test organisms treated with metal oxide nanoparticles and nanoparticle combinations are shown in Figure 7. The microscopic images corroborated with the flow cytometric analysis. 

## 3. Discussion

Gram-positive and Gram-negative bacteria differ mainly in their cell wall and cell membrane makeup. The thick cell wall of Gram-positive bacteria is made of several layers of peptidoglycan and teichoic acids, while Gram-negative bacteria have a comparatively thin cell wall with an outer membrane composed of lipopolysaccharides and lipoprotein bilayers [22]. In contrast to Gram-negative bacteria, the strong cellular wall of Gram-positive bacteria may inhibit the uptake of nanoparticle-encapsulated drugs [23]. Fungal cell walls are more protective than bacterial cell walls as they have an outer and inner cell wall composed of chitin, glycoproteins, and glucans, and hence antifungal activity could be much more complex than antibacterial [24]. These protective layers prevent the entry of antimicrobials, rendering them resistant. Metal oxide nanoparticles, however, have been demonstrated to possess a better antimicrobial activity, though their exact mechanism is not well understood [25]. Many researchers have proposed a few mechanisms, such as (i) the generation of reactive oxygen species (ROS), (ii) damage to the cellular integrity due to the interaction between metal nanoparticles and the bacterial/fungal cell walls, (iii) the release of metal ions from the nanoparticles, and (iv) internalization of the nanoparticles by the bacterial/fungal cells [26]. The electrostatic attraction between the positively charged nanoparticles and the negatively charged bacterial/fungal cell wall facilitates their attachment and penetration into the cell membrane [27]. The increased release of metal ions from the metal oxide nanoparticles post-internalization triggers an enhanced ROS generation, which leads to increased lipid peroxidation and eventually cell disruption [28]. Though there are previous reports on the antibacterial and antifungal activity of CuO and ZnO nanoparticles against bacterial and fungal species, very little is known regarding the antimicrobial activity of WO_3_ nanoparticles.

There are several reports on the antimicrobial potentials of metal nanoparticles synthesized using biological or non-biological methods [29,30,31,32]. However, it is important to explore more biogenically synthesized metal nanoparticles that are capable of destroying potent pathogens. The present study highlights the biogenic synthesis and application of significantly lower concentrations (ng/mL) of metal nanoparticles to kill the clinical pathogens against the concentrations (µg/mL) reported in the literature (Table 3). The MIC of chemically synthesized ZnO nanoparticles against *S. aureus* and *E. coli* was reported as 3.9 and 7.81 µg/mL respectively [29], while the MIC for chemically synthesized CuO has been reported as 100 µg/mL against *E. coli* [31]. Biosynthesized CuO nanoparticles have been reported at 11.6 µg/mL against *S. aureus* [33]. Chemically synthesized WO_3_ nanoparticles have been reported to exhibit MIC of 100 µg/mL [32]. In comparison to the concentrations mentioned in the literature, the biogenic nanoparticles synthesized in the current study exhibited a multifold lower MIC against all clinical pathogens. The surface area to volume ratio is a major contributor to the activity of nanoparticles, whereby the smaller the nanoparticles, the better their activity. The average particle size of chemically synthesized CuO and ZnO nanoparticles has been reported as 34.4 nm [34] and 27.49 nm [35], respectively, as determined by XRD, while the WO_3_ nanoparticles ranged from 32 nm for nanodots to 2 µm for microcrystals [36]. The commercially available CuO, ZnO, and WO_3_ nanoparticles from Sigma-Aldrich also exhibited a particle size of <50 nm for CuO and <100 nm for ZnO nanoparticles analyzed by TEM, respectively [37,38]. Few researchers have reported a wide size range of chemically synthesized nanoparticles, some having been smaller than the current biosynthesized nanoparticles. Azam et. al. (2012) [39] have reported the size-dependent antimicrobial activity of the chemically synthesized CuO nanoparticles, where the small-sized nanoparticles exhibited better activity than the large-sized particles, as determined by the zone of inhibition [39]. Hence, overall, the biosynthesized nanoparticles demonstrate a better potential than the chemically synthesized metal oxide nanoparticles. The biogenic metal oxide nanoparticles synthesized using the bacterium *S. maltophilia* possessed an average particle size between 38–53 nm, and were thus similar in size to the chemically synthesized nanoparticles but exhibited better antimicrobial properties. 

In this study, we have demonstrated the antimicrobial effect of metal oxide nanoparticles in single and in combination, and also deduced their possible mechanism of action. The MIC of the three nanoparticles against the five selected test organisms was established using the resazurin dye reduction test. CuO nanoparticles were the most effective against all test organisms under dark incubation, while WO_3_ nanoparticles were the least effective. However, upon photoactivation, the ability of WO_3_ nanoparticles to inhibit the test organisms surpassed both the CuO and ZnO nanoparticles. This signifies the photocatalytic effect of the biogenic WO_3_ nanoparticles in destroying the microbial cells, as suggested by Bamwenda and Arakawa (2001) [40] for tungsten nanoparticles synthesized by air annealing. The microbicidal effect of the three metal nanoparticles against the test pathogens was in agreement with the observed MIC results, except for the ZnO nanoparticles, which surprisingly exhibited a surge in their ability as an anti-microbicidal agent upon photoactivation [33].

ROS harm bacterial cells by damaging several cellular components. These are generated endogenously during mitochondrial respiration [41]. Though formed by the cell, ROS are neutralized by enzymes including superoxide dismutases and peroxides. However, excess production of ROS can also be observed due to the radiation and metabolism of drugs or xenobiotics [40], which is beyond the enzymes capacity to neutralize. A major effect of ROS is lipid peroxidation, which damages the phospholipid-rich bacterial cell membranes leading to cell leakage and death [42,43]. Bacterial cells incorporate linoleic acids and phospholipids in their membrane, which during lipid peroxidation, results in the formation of lipid hydroperoxide, which is further converted to aldehydes including MDA in the presence of oxidation products [41]. MDA formation is hence used as a marker to study the damage to cell membranes [44]. Thus, there is a positive correlation between ROS generation and MDA production, as increased ROS generation leads to increased lipid peroxidation, resulting in enhanced MDA levels in the cell [13]. The damage to the cellular membrane integrity of the clinical pathogens due to actions of the CuO, ZnO, and WO_3_ nanoparticles was demonstrated by the enhanced MDA production (due to lipid peroxidation) and live/dead staining using differential stains. All the three nanoparticles in singles and in combinations exhibited an increased production in MDA, with the combinations producing significantly higher amounts than the single nanoparticles. This establishes the synergistic effect of the metal nanoparticles as antimicrobials. C2 proved to be the best anti-microbial combination after photoactivation, as it exhibited almost similar levels of MDA production amongst the five test organisms, while C3 worked well in the dark. Damage to the intact cell wall and membrane was evident from the live/dead staining visualized using fluorescence microscopy and flow cytometry. CuO nanoparticles were the most efficient in damaging the cells of all selected test organisms under dark treatment, while WO_3_ nanoparticles were found to be the most effective against all test organisms with more than 70% of the damaged population under light. These results thus confirm the mechanism of antimicrobial action through ROS generation and lipid peroxidation. 

A photocatalyst is considered efficient if its band-gap energy is smaller than 3 eV, which extends the light absorption in the visible region thereby encouraging the possible use of solar energy. WO_3_ nanoparticles possess a band gap energy of 2.6–2.8 eV, and hence can be activated by visible light. Other metal oxide nanoparticles possess a high band-gap energy and require UV irradiation for photoactivation, which thereby limits their application as a photocatalyst [21,45]. Photoactivation of WO_3_ nanoparticles results in the generation of electron/hole pairs which react with water and oxygen to form superoxide anion radicals (·O_2_^−^) and hydroxyl radicals (·OH), which are highly reactive, and considered as a major contributor towards the death of microbial cells [46,47,48,49]. 

CuO and ZnO nanoparticles did not exhibit any notable changes in their antimicrobial activity upon the induction of visible light as previously reported for chemically synthesized ZnO nanoparticles [50]. Electronic defects in the CuO and ZnO nanoparticles exhibit a key role in electron-hole pair production and thereby the generation of ROS [51,52]. Several reports have mentioned the mechanism of the antibacterial activity of CuO and ZnO in the dark due to superoxide (·O_2_^–^) radical-mediated generation of ROS through singly ionized oxygen vacancy in the nanoparticle, unlike other metal nanoparticles which release metal ions [47,51]. 

The concoctions of nanoparticles induced with light have resulted in maximum cell death of all the test organisms. This could be due to the combined effect of the nanoparticles. The photoactivated WO_3_ nanoparticles induce ROS generation which results in the disruption of the cells. This activity is enhanced by the presence of CuO and ZnO nanoparticles leading to maximum cell death [52,53,54].

In the case of tungsten trioxide nanoparticles, the reduction in microbial pathogens can be attributed to photoactivation, in which the reactive species disrupt the cell membrane, thereby enabling the internal components to leak out, and ultimately the death of microbe. The synergistic action of these nanoparticles can be further explored for developing commercial antimicrobials in the form of ointments for topical applications. 

## 4. Materials and Methods

### 4.1. Synthesis of CuO, ZnO, and WO_3_ Nanoparticle

CuO, ZnO, and WO_3_ nanoparticles were synthesized in the laboratory using a bacterium, *Stenotrophomonas maltophilia*. The studies were conducted in Erlenmeyer flasks containing sterile Luria-Bertani broth. The broth was inoculated with 1 mL of 18 h old culture of *S. maltophilia* (O.D 0.8 at 600 nm) and incubated at 30 °C at 150 rpm for 24 h to allow the culture to grow. CuSO_4_, ZnSO_4_, or Na_2_WO_4_ (Sigma Aldrich, St. Louis, MO, USA) stock solutions were added to three culture flasks, respectively, to attain a 2 mM concentration, while the pH of the media was adjusted to 8 using 1 M NaOH. The flasks were further incubated at 40 °C for three days at 200 rpm. After incubation, the cells were lyzed using a QSONICA, Germany, sonicator with an amplitude of 40% and a pulse of 15 s for 10 min under ice-cold conditions, and the cell lysate was then centrifuged at 4000 rpm for 5 min. The supernatant was vacuum filtered through a 0.22 μm cellulose acetate membrane, and the filtrate was dialyzed using a snakeskin^TM^ dialysis membrane (10 K MWCO), Thermoscientific^TM^, Waltham, U.S. The dialyzed samples were lyophilized to obtain powdered CuO, ZnO, and WO_3_ nanoparticles, respectively. The nanoparticles were characterized using UV–Visible spectroscopy, particle size analysis, X-ray diffraction, transmission and scanning electron microscopy analysis, and energy dispersive X-ray spectroscopy. The lyophilized samples of the nanoparticles were also analyzed for X-Ray Diffraction on a Bruker AXS Kappa APEX II CCD X-ray diffractometer Karlsruhe, Germany, operated at 40 kV and 40 mA with Cu Kα radiation (1.54 Å) as a source. A continuous scan mode was applied with a step width of 0.020, a sampling time of 57.3 s, and a measurement temperature of 25 °C. The average grain size of the CuO, WO_3_, and ZnO nanoparticles was calculated from the full width at half maximum (FWHM) of the diffraction curves using the Debye–Scherrer formula [55].
$$D=\frac{k\lambda }{\beta cos\;\theta }$$

The scanning range of 2*θ* was between 3° and 80°, respectively. The morphology and size of the individual nanoparticles were analyzed from the images obtained from JEOL JSM-7600F FEG-SEM, Japan. FTIR analysis was performed using the ATR-FTIR Shimadzu, Japan. IRSpirit to confirm the presence of capping agents in microbially synthesized nanoparticles [17,18,19].

### 4.2. Collection of Clinical Isolates

Clinical laboratory isolates from skin, stool, oral, and vaginal swabs from the RAK Hospital, Ras Al-Khaimah, United Arab Emirates were collected for the study. Clinical samples were collected from the patients prior to administration of any antibiotic treatment. Sterile containers and swabs were used for sample collection and the samples were immediately transferred to sterile screw-capped bottles containing the transport medium. Samples were cultured on different media for the isolation of potential pathogens. Stool samples were cultured on sterile Mac Conkey’s agar, skin swab on sterile Luria-Bertani agar, and oral and vaginal swabs on sterile Sabouraud dextrose Agar, respectively. All the dehydrated media used in the study were procured from Sigma-Aldrich, St. Louis, MO, USA. *Staphylococcus aureus* and MRSA strains were isolated from skin samples, *E. coli* from stool samples, and two isolates of *Candida albicans* were successfully isolated from oral and vaginal samples, respectively. The isolates were identified using the BD Phoenix M50 (Becton, Dickinson and Company, Sparks, MD, USA) automated system for microbial identification and antibiotic susceptibility testing. 

The isolates exhibited resistance to the routinely used antibiotics, where S. *aureus* was resistant to 1 µg oxacillin; MRSA to 1 µg oxacillin, 30 µg cefoxitin, 5 µg ciprofloxacin, and 10 µg ampicillin, Thermoscientific, USA; *E. coli* was resistant to 30 µg ceftazidime, 30 µg cefotaxime, 30 µg ceftriaxone, and 30 µg aztreonam, while both isolates of *Candida albicans* from oral and vaginal samples were sensitive to all the tested antifungal agents. 

### 4.3. Minimum Inhibitory Concentration by Resazurin Assay

The five microbial isolates, *Staphylococcus aureus*, MRSA, *E. coli*, *Candida albicans* (O) (from oral samples) and *Candida albicans* (V) (from vaginal samples) from clinical samples were maintained in sterile Mueller–Hinton agar (Sigma-Aldrich, St. Louis, MO, USA) and were inoculated in Muller–Hinton media (Sigma-Aldrich, St. Louis, MO, USA) to reach to 0.8 OD at 600 nm.

The minimum inhibitory concentration (MICs) of CuO, ZnO, and WO_3_, nanoparticles were determined with the resazurin microtiter assay. Resazurin (Sigma, USA) is a non-toxic, non-fluorescent blue dye that has been widely used to assess cell viability. The metabolically active cells of the bacteria reduce blue resazurin to pink resorufin, which is further reduced to a colorless hydroresorufin by oxidoreductase within the viable cells. The change in color is distinctly visible, thus eliminating the use of the spectrophotometer [56].

CuO, ZnO, and WO_3_ nanoparticles were added into the first row of a 96-well micro titer plate containing 100 μL of Luria-Bertani broth to achieve a concentration of 100 ng/μL. Next, 1:2 dilutions of the nanoparticles were made with the Luria-Bertani broth till a concentration of 7.8 ng/μL was obtained. A total of 100 μL of the microbial culture was added to the wells to attain a final concentration of 10^6^ cells/mL. The experiment was designed with two sets of controls for microbial activity—a negative control where the microbial culture was replaced with saline, and a positive control where the nanoparticles were replaced with saline. The entire experimental setup was duplicated to incubate one set in the dark, while the other was incubated in the presence of white light from a 300 W tungsten source with a visible light wavelength range (400–700 nm) for 24 h at 37 °C. After incubation, the titer plates were stained with 10 μL of resazurin (0.05%) and further incubated in the dark at 37 °C for 2 h, following which they were observed for color change. The change in color from blue to pink revealed the reduction of resazurin to resorufin which confirmed the presence of active bacteria. All assays were performed in triplicates, and the minimum concentration of nanoparticles exhibiting no color change was considered as the MIC of the nanoparticles towards that organism.

### 4.4. Minimal Microbicidal Activity

The contents of all the wells exhibiting a negative result in the resazurin test were streaked on sterile Muller–Hinton agar (MHA) to determine the viability of the cells. The lowest concentration of the nanoparticles demonstrating a loss of microbial viability was considered as the minimal microbicidal concentration (MMC) [57]. 

### 4.5. Combination of Nanoparticle Formulation

Based on the MIC values of the individual nanoparticles, a concoction of the three nanoparticles was developed. The lowest MIC demonstrated by a specific nanoparticle against either of the selected five organisms was considered for the concoction development. The MIC of the nanoparticle was considered as full-strength, and half- and quarter-strengths were made to achieve 3 levels of nanoparticle concentration. A combination of the three nanoparticles was formulated with different levels of nanoparticle concentration to make the varied concoctions as listed in Table 4. 

### 4.6. Determination of Malondialdehyde (MDA)

Malondialdehyde (MDA) is a natural by-product of lipid peroxidation of polyunsaturated fatty acids caused by reactive oxygen species (ROS) production, which commonly serves as a marker for oxidative stress [58]. Lipid peroxidation was evaluated using the thiobarbituric acid reactive substances assay (TBARS) OxiSelect™ TBARS Assay Kit (Cell Biolabs, San Diego, CA, USA). The microbial cultures were washed thrice with sterile Dulbecco’s phosphate buffered saline (Sigma Aldrich), and the cell density was adjusted to 10^6^ cells/mL. The microbial cultures were treated with individual CuO, ZnO, and WO_3_ nanoparticles at their MIC concentrations and by the three combinations mentioned in Table 4. Each treatment was designed in two sets, one incubated in the dark and another in white light for 24 h. TBARS assay reagents were added to each experimental tube as per the protocol mentioned in the kit and were then further incubated at 95 °C. for 50 min. The assay tubes were cooled to room temperature and centrifuged at 3000× *g* for 15 min. The supernatants were then collected, and the absorbance was read at 532 nm. The concentration of MDA formed in each experimental tube was determined using the standard calibration curve established with MDA. All assays were performed in triplicates. 

### 4.7. Flow Cytometry Analysis of Dead/Live Microbial Cells and Membrane Integrity Assay by Fluoresce Microscopy Imaging

Overnight grown cultures of all five test organisms were treated with individual nanoparticles at their MIC and in combinations as designed in Table 4. A negative control with only media was maintained for each experimental set to standardize/calibrate the flow cytometer, while a positive control of untreated microbial cells was maintained to determine the maximum cell viability of each test organism. All experiments were performed in triplicates while maintaining two sets, one set incubated in the dark while the other set incubated in the presence of light at 37 °C. for 24 h. The cells were washed thrice with sterile Dulbecco’s phosphate buffer saline (Sigma Aldrich) and resuspended in sterile Dulbecco’s phosphate buffer saline at a cell density of 10^6^ cells/mL for the analysis.

Flow cytometry was performed to analyze the cell viability of the microbes using the LIVE/DEAD BacLight cell Viability kit (ThermoFisher, Waltham, MA, USA) which uses propidium iodide (PI) and SYTO^®^ 9 fluorescent dyes to differentiate between living and dead cell populations [59]. The SYTO 9 is a green-fluorescent nucleic stain that typically stains all microbial cells in a population, including those with intact membranes and those with damaged membranes. Propidium iodide, on the other hand, is a red fluorescent intercalating stain that is impermeable to microbial cells with intact cell membranes owing to its huge molecular size. Thus, bacteria with broken membranes will appear fluorescent red, while bacteria with intact cell membranes will appear fluorescent green [60].

A staining solution containing both dyes was prepared from the stock solutions provided in the BacLight cell Viability kit as per the protocol mentioned by the manufacturer. The staining solution was added to 1:1000 diluted microbial cultures in the experimental tubes and incubated in the dark at room temperature for 15 min. Following incubation, the samples were analyzed using a calibrated BD FACSAria III Flow cytometer using BD FACSDiva software. The excitation optics were set to 488 nm from a blue solid-state laser at 50 mW, while optical filters were set to measure green fluorescence (SYTO^®^ 9) at 520 nm (FL1), and red fluorescence (propidium iodide) above 630 nm (FL3). The trigger was set for the green fluorescence channel FL1. Positive control, negative control, and fluorescence minus one (FMO) controls were used to set the appropriate gates. A total of 10,000 events were recorded, and the acquisition gate for the positive control was established using forward scatter and side scatter channels to eliminate the background noise and debris from the sample.

All samples in the experimental tubes were also observed under an Olympus BX53 microscope (Olympus, Tokyo, Japan), with epifluorescence attachment to study the cell morphology and image the live and dead cells. The samples were excited at 485 nm, while the emission filters selected were 500 nm and 635 nm, respectively. 

The live cells with intact cell membranes appeared green colored at 500 nm due to the nuclear dye SYTO^®^ 9, while the dead cells stained red due to the penetration of propidium iodide, which was measured at 635 nm with an excitation wavelength of 485 nm [61].

### 4.8. Statistical Analysis

All analyzes were performed using origin pro 2019 and RStudio (v 1.0.136, Boston, MA, USA), software with graphics coded via the Ggplot2 package. Experimental data were statistically evaluated by the one-way analysis of variance (ANOVA). A Tukey’s post hoc test was performed to identify the distinct groups that significantly differed from one another across different dependent variables. All statistical interpretations were based on a 5% probability (*p* < 0.05) [62,63,64]. 

## 5. Conclusions

In summary, light-induced WO_3_ nanoparticles were found to be much more effective at lower concentrations than CuO and ZnO nanoparticles against Gram-negative *E. coli*, Gram-positive *S. aureus* and MRSA, and eukaryotic *C. albicans* (O) and *C. albicans* (V). However, the metal nanoparticles were much more effective (>90% antimicrobial activity) at very low concentrations when used in combinations as compared to when applied individually. This study demonstrates the potential for a light-induced synergistic effect of WO_3_ nanoparticles in combination with CuO and ZnO nanoparticles as compared to single application, as well as the ability of the nanoparticle combination to target a broader spectrum of microbes, including Gram-positive, Gram-negative, and fungal pathogens. This research may help and encourage the engineers and medical professionals in the design and development of medical/pharmaceutical devices and instruments coated with nanoparticles which can be photo-sterilized, reducing the time and cost of sanitization which target a broad spectrum of microbial infections.

## Figures and Tables

**Figure 1 ijms-24-09998-f001:**
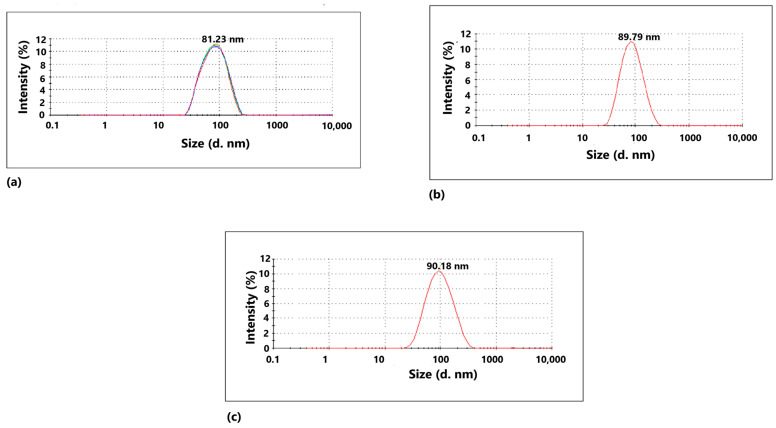
Particle size distribution of (**a**) CuO, (**b**) ZnO, and (**c**) WO_3_ nanoparticles, which were determined using a Malvern Zetasizer Nano DS analyzer. All nanoparticles were suspended in HPLC grade water.

**Figure 2 ijms-24-09998-f002:**
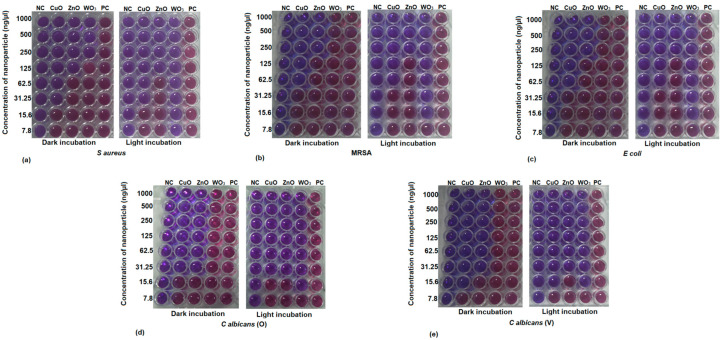
Resazurin assay for identifying the minimum inhibitory concentration for CuO, ZnO, and WO_3_ nanoparticles against (**a**) *S. aureus* (**b**) MRSA, (**c**) *E. coli*, (**d**) *C. albicans* (O), and (**e**) *C. albicans* (V). NC is the negative control where microbial culture was replaced with saline, while PC is the positive control where nanoparticles were replaced with saline.

**Figure 3 ijms-24-09998-f003:**
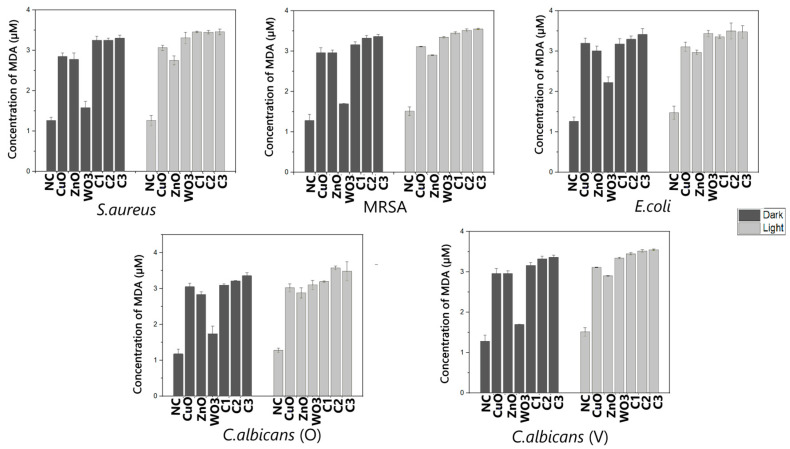
Quantification of MDA production in *S. aureus*, MRSA, *E. coli*, *C. albicans* (O), and *C. albicans* (V) after 24 h treatment with the MICs of CuO, ZnO, and WO_3_ nanoparticles in singles and in the combinations C1, C2, and C3 under light and dark incubation. NC is the is the negative control where the nanoparticles were replaced with saline. Results are expressed as mean ± SD (n = 3).

**Figure 4 ijms-24-09998-f004:**
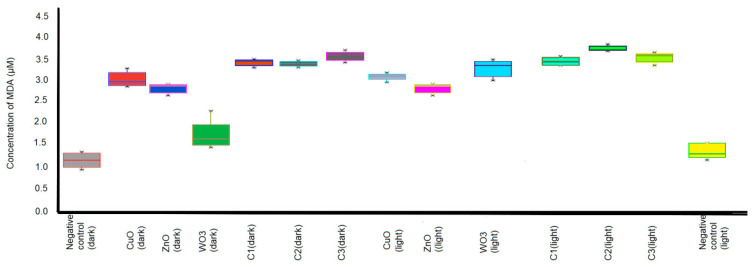
Boxplot demonstrating the mean concentration of MDA produced by all the test organisms when treated with CuO, ZnO, and WO_3_ nanoparticles in singles and in the combinations C1, C2, and C3 under light and dark incubation. The top and bottom boundaries of each box indicate the upper and lower quartile values, and the horizontal lines inside each box represent the median. The ends of the whiskers mark the lowest and highest MDA production observed from each treatment irrespective of culture.

**Figure 5 ijms-24-09998-f005:**
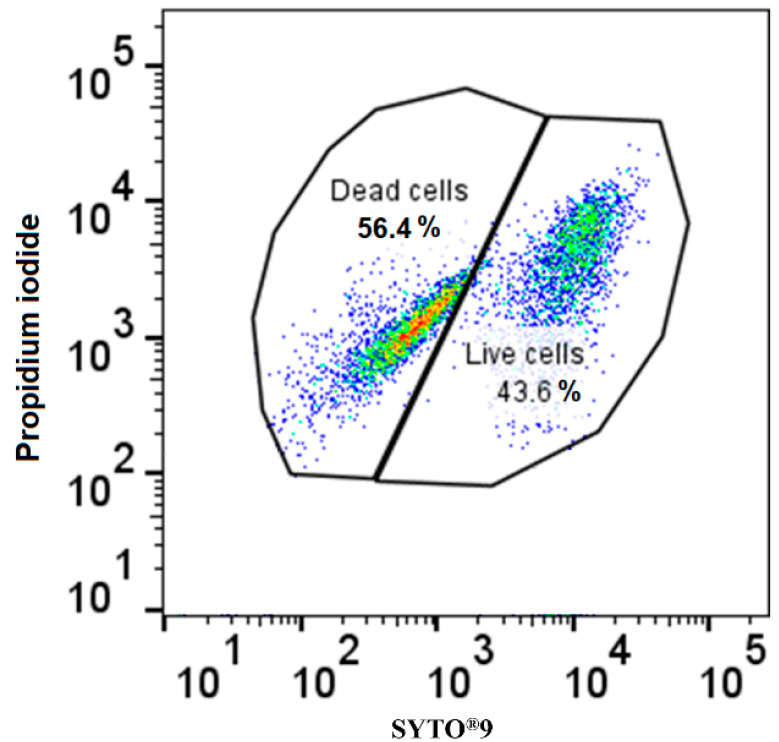
Gating strategy control using *E. coli* suspension comprising equal proportions of live and dead (heat killed) cells using the BD FACSAria III flow cytometer and FlowJo software v10.

**Figure 6 ijms-24-09998-f006:**
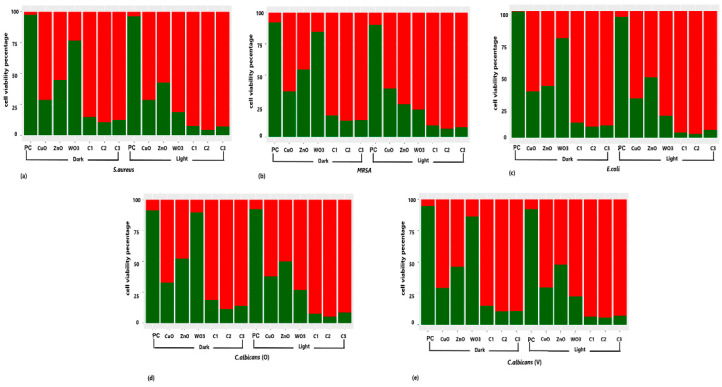
The effect of CuO, ZnO, and WO_3_ nanoparticles in singles and in the combinations C1, C2, and C3 under light and dark incubation conditions on (**a**) *S. aureus*, (**b**) MRSA, (**c**) *E. coli*, (**d**) *C. albicans* (O), and (**e**) *C. albicans* (V), as studied using flow cytometric analysis. Positive control (PC) demonstrates the maximum viability expressed by the untreated cells of the test organisms. Green bars represent the live population, while red bars represent the population of dead cells (n = 3).

**Figure 7 ijms-24-09998-f007:**
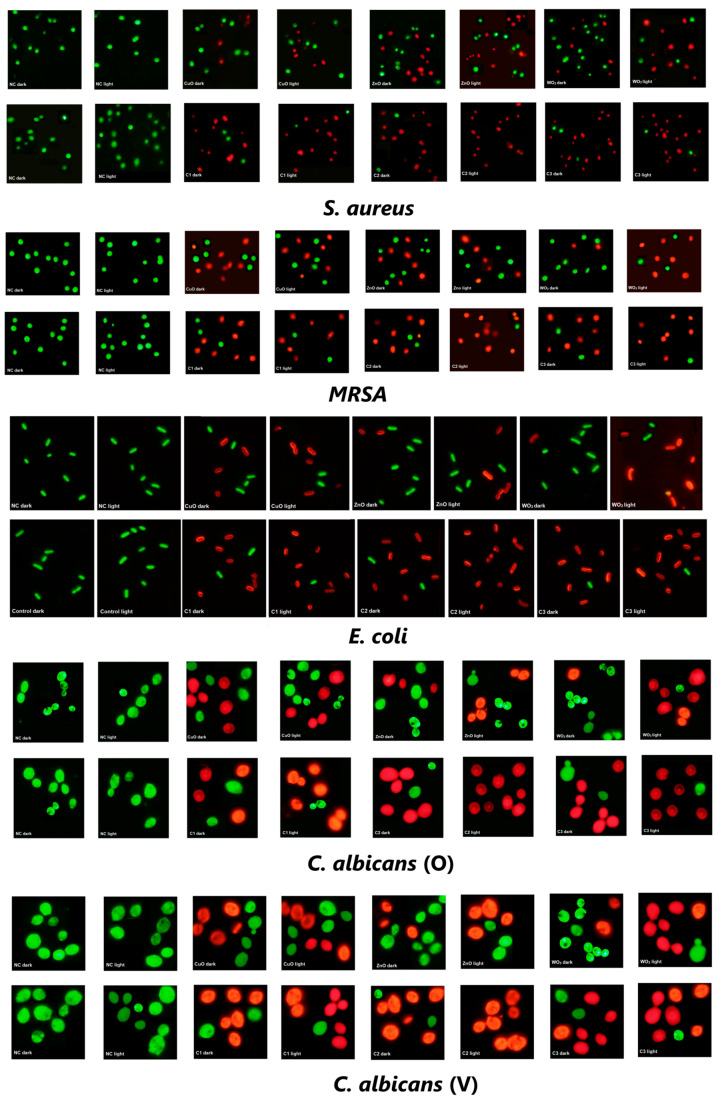
The fluorescence microscopy images of *S. aureus*, *MRSA*, *E. coli*, *C. albicans* (O) and *C. albicans* (V) under 40× magnification, after 24 h exposure to the MIC of CuO, ZnO, and WO_3_ nanoparticles in singles and in the combinations C1, C2, and C3 under light and dark incubation. The clinical isolates were stained with BacLight kit. Live cells are stained green with SYTO9 while dead cells stain red with PI.

**Table 1 ijms-24-09998-t001:** Determination of the crystallite size of the biosynthesized CuO and ZnO nanoparticles, synthesized using XRD analysis.

	Peak Position	FWHM	Crystallite Size (nm)	Average Grain Size (nm)
CuO	20.90853	0.22918	35.24957	38.96 (±5)
	35.37912	0.27172	30.68807	
	38.61757	0.1803	46.68871	
	48.62369	0.27196	32.05383	
	61.41574	0.25925	35.64097	
	66.0172	0.17732	53.42576	
WO_3_	23.19039	0.18668	43.44222	53.03 (±5)
	23.67589	0.20875	38.88346	
	24.41808	0.19304	42.10592	
	31.77072	0.1311	63.00318	
	45.52266	0.18972	45.41009	
	66.26026	0.09757	97.22758	
	56.53233	0.21911	41.16581	
ZnO	23.28048	0.26135	31.03541	42.64 (±3)
	31.8997	0.24101	34.28227	
	33.47986	0.21567	38.46548	
	34.76525	0.17377	47.90504	
	50.91306	0.14305	61.50663	

**Table 2 ijms-24-09998-t002:** Minimum inhibitory concentration (MIC) and minimum microbicidal concentration (MMC) of CuO, ZnO, and WO_3_ nanoparticles against the test microorganisms, *S. aureus*, methicillin-resistant *S. aureus*, *E. coli*, *C. albicans* (O), and *C. albicans* (V).

Clinical Isolate	Incubation in Dark						Incubation in Light					
	CuO		ZnO		WO_3_		CuO		ZnO		WO_3_	
	Concentration of Nanoparticles (ng/µL)						Concentration of Nanoparticles (ng/µL)					
	MIC	MMC	MIC	MMC	MIC	MMC	MIC	MMC	MIC	MMC	MIC	MMC
*S. aureus*	31.25	62.5	125	250	250	-	31.25	62.5	125	125	<7.8	7.8
*MRSA*	62.5	125	250	250	-	-	62.5	125	250	250	15.6	15.6
*E. coli*	62.5	62.5	250	250	-	-	62.5	62.5	250	31.25	15.6	7.8
*C. albicans (O)*	31.25	125	31.25	62.5	-	-	31.25	125	31.25	62.5	15.6	15.6
*C. albicans (V)*	15.6	31.25	31.25	62.5	-	-	15.6	31.25	31.25	62.5	15.6	15.6

**Table 3 ijms-24-09998-t003:** A comparison between the minimal inhibitory concentrations of the CuO, ZnO, and WO_3_ nanoparticles synthesized by *S. maltophilia* and those reported in literature.

Nanoparticle	Test Organism	Minimal Inhibitory Concentration in Literature	Minimal Inhibitory Concentration of *S. maltophilia* Synthesized Nanoparticles
ZnO (chemically synthesized)	*S. aureus*	3.9 µg/mL [39]	31.25 ng/mL
	*E. coli*	7.81 µg/mL [39]	62.5 ng/mL
CuO (biogenic)	*S. aureus*	11.6 µg/mL [40]	31.25 ng/mL
CuO (chemically synthesized)	*E. coli*	100 µg/mL [30]	62.5 ng/mL
WO_3_ (chemically synthesized)	*S. aureus*	100 µg/mL [41]	7.8 ng/mL

**Table 4 ijms-24-09998-t004:** Combination of the metal oxide nanoparticles to formulate different concentrations of concoctions to study the antimicrobial activity.

Nanoparticle Combinations	Concentration (ng/µL)		
	CuO	WO_3_	ZnO
Combination 1 (C1)	15.6	1.95	15.6
Combination 2 (C2)	7.8	7.8	7.8
Combination 3 (C3)	3.9	3.9	31.25

## Data Availability

The data presented in this study are available in this article and the Appendix A.

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
