# Peer review of "Antimicrobial Activity of Biogenic Metal Oxide Nanoparticles and Their Synergistic Effect on Clinical Pathogens"

_ijms, 2023, doi:10.3390/ijms24129998_

Round 1

Reviewer 1 Report

The manuscript entitled “Antimicrobial activity of photocatalytic metal oxide nanoparticles and their synergistic effect on clinical pathogens” by Dali Vilma Francis, Manju Nidagodu Jayakumar, Hafiz Ahmad, Trupti Gokhalen is quite interesting and presents interesting results about the antimicrobial activity of nanoparticles and their effect combining UV light. Is importat that the mechanism was explored using different techniques for a possible pathway establishment. Some points must be addressed by the authors before its publication.

1.       The Introduction must include a justification for the usage of the three different compositions of nanoparticles.

2.       In “Materials and methods” section, the protocol for nanoparticles synthesis must be described (reagents, equipment, time of reaction, protocol for particles processing) or adequate citation must be included.

3.       Equipment (brand, model) and samples preparation for XRD analysis were not reported. Any diffractogram were included in the manuscript, please include the analysis.

4.       Size and shape of nanoparticles is needed to stablish correlations about the effect of nanoparticles. The authors must provide these properties.

5.       Revise Journal´s guidelines for figures.

Minor revision about grammar and readibility

Author Response

The manuscript entitled “Antimicrobial activity of photocatalytic metal oxide nanoparticles and their synergistic effect on clinical pathogens” by Dali Vilma Francis, Manju Nidagodu Jayakumar, Hafiz Ahmad, Trupti Gokhalen is quite interesting and presents interesting results about the antimicrobial activity of nanoparticles and their effect combining UV light. Is importat that the mechanism was explored using different techniques for a possible pathway establishment. Some points must be addressed by the authors before its publication.

  1. The Introduction must include a justification for the usage of the three different compositions of nanoparticles.

Justification for the use of copper, zinc and tungsten nanoparticles has been added.

  1. In “Materials and methods” section, the protocol for nanoparticles synthesis must be described (reagents, equipment, time of reaction, protocol for particles processing) or adequate citation must be included.

The detailed protocol on nanoparticle synthesis has been added.

  1. Equipment (brand, model) and samples preparation for XRD analysis were not reported. Any diffractogram were included in the manuscript, please include the analysis.

The details on characterization of nanoparticles using XRD analysis and other techniques such as FTIR and SEM have need added. The corresponding figures are also provided in supplementary data. The details on synthesis and characterization of CuO, ZnO and WO3 have been published earlier (reference 16, 17, 18) by the authors and hence we had avoided the repetition.

  1. Size and shape of nanoparticles is needed to stablish correlations about the effect of nanoparticles. The authors must provide these properties.

The Figure S3 demonstrated the size and shape of the CuO, ZnO and WO3 nanoparticles.

  1. Revise Journal´s guidelines for figures.

The figures have been revised as per the journals guidelines.

Reviewer 2 Report

Please find attached the review.

Please find attached the review. English should be approved.

Author Response

Reviewer 2

Manuscript entitled "ANTIMICROBIAL ACTIVITY OF PHOTOCATALYTIC METAL OXIDE NANOPARTICLES AND THEIR SYNERGISTIC EFFECT ON CLINICAL PATHOGENS" is very intersting.

Authors found that CuO and ZnO nanoparticles exhibited good antimicrobial activity under dark incubation which did not alter on photoactivation.

In the abstract is written “good antimicrobial activity under dark incubation”, can you change good with significant, or replace the word “good” with percentage?

The authors agree on the suggestion and have replaced the word ‘good’ with ‘significant’.

They also found that, photoactivated WO3 nanoparticles “significantly reduced the number of viable cells of all the test organisms”. Can you add percentage of this reducing in the abstract?

The authors agree on the suggestion and have reframed the sentence to: ‘However, photoactivated WO3 nanoparticles significantly reduced the number of viable cells by 75% -for all the test organisms thus proving to be a promising antimicrobial agent.’  

“Combinations of metal nanoparticles demonstrated synergistic action as a significant increase in their antimicrobial property was observed compared to action of single elemental nanoparticles.” Can you write what are the combinations, in the abstract?

The authors agree on the suggestion and have reframed the sentence to: ‘Combinations of CuO, ZnO and WO3 nanoparticles demonstrated synergistic action as a significant increase in their antimicrobial property (>90%) was observed compared to action of single elemental nanoparticles.’

English grammar and spelling can be approved. For example, on page 4, the sentence "Random combination of the three nanoparticles were prepared at different strengths to make the varied concentration concoctions are lister in Table 1...."

The authors apologize and have corrected the spelling mistakes.

Reviewer 3 Report

The article entitled “Antimicrobial activity of photocatalytic metal oxide nanoparticles and their synergistic effect on clinical pathogens” by Trupti Gokhale and co-workers reports the action of CuO, ZnO, and WO3 nanoparticles and of their combinations towards Clinical isolates of E. coli, S. aureus, methicillin resistant S. aureus (MRSA), and Candida albicans from oral and vaginal samples incubated under dark and light conditions.  The mechanism of the antimicrobial action was evaluated by measuring malondialdehyde (MDA) production and the damage to cell integrity using live/dead staining and quantitating by flow cytometry.

Although the subject has interest there are some questions  that need to be answered and aspects clarified: 

i) As the authors know there are a lot of works concerning the action of ZnO and CuO as antimicrobial agents. It is not clear the novelty of the nanoparticles used? These biogenic nanoparticles have better performance than the ones described in the literature and prepared using conventional procedures?  A comparative evaluation should be done or at least some comparison with results from literature.

ii) Some information concerning the synthesis and characterization of the nanoparticles must be introduced in SI although the preparation is not new.

iii) Another question is related with the formation Malondialdehyde that the authors justify as lipid peroxidation due to ROS (reactive oxygen species). How the waters envisage the formation of those ROS under dark conditions and light conditions?  Are the authors able to identify which are the most important ones? Clarify the mechanism of the formation of malondialdehyde during the peroxidation process.

iv) Clarify how the  combination of nanoparticle formulation was performed and what was the positive effect of these formulations compared with the isolate nanoparticles. How the authors justify their good  performance  under dark conditions? Probably a characterization of those nanoparticles must be performed.

v)   In Fig 1 indicate what is  NC and PC

vi) Improve the quality of Table 2 by reducing the size of the abbreviations. It is difficult to read.

vii)    In the section BD FACSAria III flow cytometer and FlowJO software section indicate also the reductions in terms of log.

viii)  Correct the structures of superoxide anion radicals and hydroxyl radicals (OH-)

xi)      The title must be revised since the Antimicrobial activity of some metal oxide nanoparticles does not require light.

Minor editing of English language is  required

Author Response

  1. i) As the authors know there are a lot of works concerning the action of ZnO and CuO as antimicrobial agents. It is not clear the novelty of the nanoparticles used? These biogenic nanoparticles have better performance than the ones described in the literature and prepared using conventional procedures?  A comparative evaluation should be done or at least some comparison with results from literature.

The authors agree with the reviewer’s suggestions and have incorporated the comparison as Table 3 in the text and discussed the under the discussion section.

  1. ii) Some information concerning the synthesis and characterization of the nanoparticles must be introduced in SI although the preparation is not new.

The detailed protocol on nanoparticle synthesis and their characterization have been added in the manuscript.

iii) Another question is related with the formation Malondialdehyde that the authors justify as lipid peroxidation due to ROS (reactive oxygen species). How the waters envisage the formation of those ROS under dark conditions and light conditions?  Are the authors able to identify which are the most important ones? Clarify the mechanism of the formation of malondialdehyde during the peroxidation process.

  1. A few sentences on the formation of ROS under dark and light conditions have been added in the discussion. In the present study, did not focus on identifying the most important ROS, but would be interested in studying the in-depth mechanism in future.

Since this work focused on sustainable alternatives to antibiotics, the authors also, did not consider photoactivation with UV light to study CuO and ZnO photoactivation.

  1. The authors have added a sentence to clarify the mechanism of formation of MDA. ‘Lipid peroxidation of bacterial cell membrane results in formation of lipid hydroperoxide which is further converted to aldehydes such as MDA which is used as a marker to study the damage to cell membranes [44].’
  2. iv) Clarify how the combination of nanoparticle formulation was performed and what was the positive effect of these formulations compared with the isolate nanoparticles. How the authors justify their good  performance  under dark conditions? Probably a characterization of those nanoparticles must be performed.

Clarify how the combination of nanoparticle formulation was performed and what was the positive effect of these formulations compared with the isolate nanoparticles

  1. To develop the nanoparticle concoction we considered the following:
  2. The lowest MIC demonstrated by a specific nanoparticle against either of the selected five organisms, was considered for the concoction development. Eg: 15.6 ng/ul was the lowest MIC for CuO against C. albicans (V) and hence was considered.
  3. While developing the concoction, the authors considered a random design to include full-strength, and half and quarter strength combination of the three nanoparticles.

Nanoparticle combinations

Concentration (ng/µL)

CuO

WO3

ZnO

Combination 1 (C1)

Full (15.6 ng/ul)

Quarter (1.95 ng/ul)

Half (15.6 ng/ul)

Combination 2 (C2)

Half (7.8 ng/ul

Full (7.8 ng/ul)

Quarter (7.8 ng/ul)

Combination 3 (C3)

Quarter (3.9 ng/ul)

Half (3.9 ng/ul)

Full (31.25 ng/ul)

  1. All nanoparticle combinations demonstrated a higher antimicrobial potential as compared to individual nanoparticles (Figure 5).

How the authors justify their good performance  under dark conditions?

All nanoparticles in single demonstrated certain level of antimicrobial activity which was estimated in terms of MDA production and loss of viability. A more than 1.5 fold increase in the MDA production was observed as compared with negative control. Changing the incubation in presence of light witnessed a more than 2 fold surge in production of MDA for isolated treated with WO3 nanoparticles. Combinations of the nanoparticles worked significantly well in dark and under light activation. Similar results were observed for the viability test using flow cytometry.

Probably a characterization of those nanoparticles must be performed.

The detailed protocol on nanoparticle synthesis and their characterization have been added in the manuscript.

v)   In Fig 1 indicate what is  NC and PC

The authors are sorry for the missing information in the legends and the following has been incorporated in the manuscript.

NC is the negative control where microbial culture was replaced with saline, while PC is the positive control where nanoparticles were replaced with saline.

  1. vi) Improve the quality of Table 2 by reducing the size of the abbreviations. It is difficult to read.

The authors have revised Table 2 as per the suggestion.

vii)    In the section BD FACSAria III flow cytometer and FlowJO software section indicate also the reductions in terms of log.

The authors have added a sentence on two log reduction of the signal observed for the live and dead populations of known test organism.

viii)  Correct the structures of superoxide anion radicals and hydroxyl radicals (OH-)

the structures have been corrected.

xi)      The title must be revised since the Antimicrobial activity of some metal oxide nanoparticles does not require light.

The authors agree with the suggestion and the title has been revised to “Antimicrobial activity of metal oxide nanoparticles and their synergistic effect on clinical pathogens”

Round 2

Reviewer 1 Report

For the revised version of the manuscript, I have some suggestions after to be consider suitable for its publication.

According with the authors, “The nanoparticles were characterized using UV–Visible spectroscopy, particle size analysis, Xray diffraction, transmission and scanning electron microscopy analysis, and energy dispersive X-ray spectroscopy. The lyophilized samples of the nanoparticles were also analyzed for X-Ray Diffraction on a Bruker AXS Kappa APEX II CCD X-ray diffractometer operated at 40 kV and 40 mA with Cu Kα radiation (1.54 Å) as a source. A continuous scan mode was applied with a step width of 0.020, sampling time of 57.3 s and measurement temperature of 25 ºC. The scanning range of 2θ was between 3_ and 80. The morphology and size of the individual nanoparticles were analyzed from the images from JEOL JSM-7600F FEG-SEM. FTIR analysis was performed using ATR-FTIR Shimadzu IR Spirit, to confirm the presence of cap-ping agents in microbial synthesized nanoparticle [17,18,19].”. However, any result regarding to particle size and UV-Vis were presented. I recommend to present the absorption spectra, the size distribution and histogram from the samples. If you have not TEM images, the size can be calculated from the SEM images.

In “Characterization of nanoparticles” section, the referred images can be part of the manuscript body´s. I suggest to include them and improve the discussion about the obtained samples. Is possible to observe intense contribution of fluorescence in the XRD patterns, besides to really low counts in the detected signal. Usually, it can be related to small particle size and broadening reflection peaks (as you presented). It can be supported by the obtained value from electron microscopy determination of size. Also, the particle sizes can be calculated by Debye-Scherrer equation using XRD data.

My suggestions are based on the fact that, better understanding about the properties of the reported nanoparticles allows to compare directly with other approaches, as you did in Table 3. As is well-known, the surface area and the reactivity are strongly dependent of particle´s size. These interactions must be related with the nanoparticle’s properties in the discussion of the Table 3 for more scientific rigor in the comparison.

minor gramatical erros detected

Author Response

According with the authors, “The nanoparticles were characterized using UV–Visible spectroscopy, particle size analysis, Xray diffraction, transmission and scanning electron microscopy analysis, and energy dispersive X-ray spectroscopy. The lyophilized samples of the nanoparticles were also analyzed for X-Ray Diffraction on a Bruker AXS Kappa APEX II CCD X-ray diffractometer operated at 40 kV and 40 mA with Cu Kα radiation (1.54 Å) as a source. A continuous scan mode was applied with a step width of 0.020, sampling time of 57.3 s and measurement temperature of 25 ºC. The scanning range of 2θ was between 3_ and 80. The morphology and size of the individual nanoparticles were analyzed from the images from JEOL JSM-7600F FEG-SEM. FTIR analysis was performed using ATR-FTIR Shimadzu IR Spirit, to confirm the presence of cap-ping agents in microbial synthesized nanoparticle [17,18,19].”. However, any result regarding to particle size and UV-Vis were presented. I recommend to present the absorption spectra, the size distribution and histogram from the samples. If you have not TEM images, the size can be calculated from the SEM images.

The authors agree with the suggestion and have included Figure 1 on Particle size distribution of CuO, ZnO and WO3 nanoparticles. We have also revised Figure S3 to include the data on particle size of the nanoparticles.

In “Characterization of nanoparticles” section, the referred images can be part of the manuscript body´s. I suggest to include them and improve the discussion about the obtained samples. Is possible to observe intense contribution of fluorescence in the XRD patterns, besides to really low counts in the detected signal. Usually, it can be related to small particle size and broadening reflection peaks (as you presented). It can be supported by the obtained value from electron microscopy determination of size. Also, the particle sizes can be calculated by Debye-Scherrer equation using XRD data.

 The authors agree with the suggestion and have included Table 2 on average particle size of all 3 metal oxide nanoparticles based on XRD analysis.

My suggestions are based on the fact that, better understanding about the properties of the reported nanoparticles allows to compare directly with other approaches, as you did in Table 3. As is well-known, the surface area and the reactivity are strongly dependent of particle´s size. These interactions must be related with the nanoparticle’s properties in the discussion of the Table 3 for more scientific rigor in the comparison.

The authors agree with the suggestion and have modified the discussion by commenting on the size of chemically synthesized nanoparticles. Few researchers have reported a wide size range of chemically synthesized nanoparticles, some smaller than the current biosynthesized nanoparticles. Azam et. al. (2012) [48] have reported the size dependent antimicrobial activity of chemically synthesized CuO nanoparticles, where smaller sized nanoparticles exhibited better activity than larger sized particles as determined by the zone of inhibition [48]. Hence, overall, the biosynthesized nanoparticles demonstrate a better potential than the chemically synthesized metal oxide nanoparticles.

Reviewer 3 Report

The authors improved the article and were able to clarify most of my concerns.  However a few revisions and simplifications  on the following  comments are required since is difficult to understand the message:

Check the phrase construction of the following comments: “Metal nanoparti[1]cles exhibit a non-specific antimicrobial activity and are could employ mechanisms such as generation of pores on the cell wall or surface membrane, ROS generation, or…..”

“A random combination of the three nanoparticles was prepared at different strengths to make the varied concentration concoctions are listed in Table 1.”

“Current study reports the application of a significantly lower concentrations (ng/ml) of biogenic metal nanoparticles to kill the clinical pathogens as against the concentrations (µg/ml) reported in literature (Table 3).”

“The combination nanoparticles induced have resulted maximum cell death irrespective of concentration or microorganisms. This could be due to the combined effect of nanoparticles, the photoactivated WO3 nanoparticles induces the ROS generation which leased to the disruption of cells which activity is enhanced by CuO and ZnO nanoparticles leading to the maximum cell death”

Correct the structure of  superoxide anion radical.

Check the references according with MDPI requirements. 

The construction of the some comments (see above) need to be rephrase and simplified 

Author Response

Check the phrase construction of the following comments: “Metal nanoparti[1]cles exhibit a non-specific antimicrobial activity and are could employ mechanisms such as generation of pores on the cell wall or surface membrane, ROS generation, or…..”

Checked, but could not identify the error.

“A random combination of the three nanoparticles was prepared at different strengths to make the varied concentration concoctions are listed in Table 1.”

The authors have revised the sentence to “A combination of the three nanoparticles was formulated with  different levels of nanoparticles concentration to make the varied concoctions as listed in Table 1.”

“Current study reports the application of a significantly lower concentrations (ng/ml) of biogenic metal nanoparticles to kill the clinical pathogens as against the concentrations (µg/ml) reported in literature (Table 3).”

The authors have revised the sentence to ‘Present study highlights the biogenic synthesis and application of significantly lower concentrations (ng/ml) of metal nanoparticles to kill the clinical pathogens as against the concentrations (µg/ml) reported in literature (Table 4).’

“The combination nanoparticles induced have resulted maximum cell death irrespective of concentration or microorganisms. This could be due to the combined effect of nanoparticles, the photoactivated WO3 nanoparticles induces the ROS generation which leased to the disruption of cells which activity is enhanced by CuO and ZnO nanoparticles leading to the maximum cell death”

The authors have revised the paragraph to “The concoctions of nanoparticles induced with light have resulted in maximum cell death of all the test organisms. This could be due to the combined effect of nanoparticles. The photoactivated WO3 nanoparticles induce ROS generation which results in disruption of cells. This activity is enhanced by presence of CuO and ZnO nanoparticles leading to the maximum cell death [55,56,57].”

Correct the structure of  superoxide anion radical.

The authors have revised the formula to (·O2)

Check the references according with MDPI requirements. 

The references are corrected as per the MDPI format.
